# Simultaneously determining the W± boson mass and parton shower model parameters

O. Lupton, M. Vesterinen

*University of Warwick, Coventry, United Kingdom*

**Abstract**

We explore the possibility of simultaneously determining the W boson mass, $m_{\mathrm{W}}$, and QCD-related nuisance parameters that affect the W boson $p_{\mathrm{T}}$ spectrum from a fit to the $p_{\mathrm{T}}$ spectrum of the muon in the leptonic decay $\mathrm{W} \to \mu\nu$. The study is performed using pseudodata generated using the parton shower event generator PYTHIA and the muon is required to fall in a kinematic region corresponding to the approximate acceptance of the LHCb detector. We find that the proposed method performs well and has little trouble disentangling variations in the muon $p_{\mathrm{T}}$ spectrum due to $m_{\mathrm{W}}$ from those due to the W boson $p_{\mathrm{T}}$ model.

# 1  Introduction

Global fits to precision electroweak observables are a powerful probe of physics beyond the Standard Model (SM). One input to these fits, the W boson mass, $m_W$, is of particular interest because it is determined indirectly by the electroweak fits more precisely than it has been measured directly. The recent Gfitter electroweak fit update [1] indirectly determines $m_W = 80.354 \pm 0.007 \, \text{GeV}/c^2$, while the latest average of direct measurements, which is dominated by inputs from the CDF [2], D0 [3] and ATLAS [4] collaborations, is $m_W = 80.379 \pm 0.012 \, \text{GeV}/c^2$ [5]. Improving the precision of the direct measurement is therefore well motivated.

Measurements of $m_W$ at hadron colliders have to date been based on three different observables in $W \to \ell\nu$ decays, where $\ell$ represents an electron or muon. These are: the transverse momentum of the charged lepton, $p_T^\ell$, the missing transverse momentum, $p_T^\nu$, and the transverse mass $m_T = \sqrt{2p_T^\ell p_T^\nu \left(1 - \cos\Delta\phi\right)}$, where $\Delta\phi$ is the opening angle between the charged and neutral lepton momenta in the plane transverse to the beams. There are two, closely related, sources of systematic uncertainty that potentially limit the precision with which $m_W$ can be measured at the LHC. The first is the parton distribution functions (PDFs) that primarily determine the rapidity, $y$, distribution of the W bosons. The second is the transverse momentum distribution of the W bosons, $p_T^W$. The $p_T^\ell$ distribution is particularly sensitive to the latter.

It has previously been suggested that a measurement of $m_W$ in the forward kinematic region covered by the LHCb experiment would be of particular interest due to the predicted anticorrelation of PDF uncertainties between measurements in the central and forward rapidity regions [6]. Further studies of the PDF uncertainties affecting an LHCb measurement of $m_W$ have been developed in Ref. [7], including suggestions of how these can be reduced by using in-situ constraints. Since the proposed LHCb measurement of $m_W$ is based on the $p_T^\ell$ spectrum, it is particularly susceptible to uncertainties in the $p_T^W$ spectrum. Our attention is therefore drawn to mitigating strategies for that source of uncertainty in the context of an LHCb measurement.

Fixed order QCD corrections to the $W^\pm$ and $Z^0$ cross sections are known fully differentially up to $\mathcal{O}\left(\alpha_s^2\right)$ [8–12], and calculations differential in the gauge boson transverse momentum, $p_T^V$, have recently been made up to $\mathcal{O}\left(\alpha_s^3\right)$ [13,14]. Electroweak corrections are known up to next-to-leading order [15–19]. These state-of-the-art fixed order calculations are crucial, and the higher order corrections are important at larger $p_T^V$ values. The bulk of the $p_T^V$ distribution is, however, situated in the $p_T^V \lesssim m_V$ region, where large logarithmic terms must be resummed to achieve an accurate prediction. This can be approached in two ways. The first is to use analytic resummation techniques, where next-to-next-to-leading logarithmic accuracy (NNLL) is well known [20–25] and N³LL [26] has recently been achieved. The second approach is to use parton-shower algorithms, such as Herwig++ [27], Pythia [28] and Sherpa [29].

While the improvements in calculations of the $p_T^W$ spectrum in recent years are impressive, the precision of the state-of-the-art calculations is yet to reach the $\mathcal{O}\left(1\%\right)$ level required for a $\mathcal{O}(10\,\text{MeV}/c^2)$ measurement of $m_W$ at the LHC. One approach to determining the $p_T^W$ spectrum with this precision is to study the $p_T^Z$ spectrum, which can be measured extremely precisely in the regions of phase space that are likely to produce two final-state leptons in the relevant detector acceptance, and use these measurements to infer the $p_T^W$ spectrum with, ideally, reduced uncertainties with respect to the direct

calculation of $p_T^W$. How best to evaluate robust theoretical uncertainties in this approach is an open topic. Independently of whether explicit constraints from $Z^0$ data are included in experimental fits of $m_W$, it is important to define well-motivated nuisance parameters that can be varied during the experimental analyses.

This paper explores the possibility of simultaneously determining $m_W$ and nuisance parameters relating to $p_T^W$ in the context of the proposed measurement of $m_W$ at LHCb using the muon transverse momentum spectrum, $p_T^\mu$. This study identifies two parameters of the PYTHIA [28] Monte Carlo generator, which strongly affect the $p_T^W$ distribution, as examples of nuisance parameters that could be varied in an $m_W$ measurement [30, 31]. One of these is related to the intrinsic parton $k_T$, and the second is related to the strong coupling constant. The ATLAS collaboration also varied the intrinsic $k_T$ cut off parameter in the AZ tune of Pythia [32], but this parameter is found to be far less influential than the two parameters that we consider. It is, of course, unlikely that PYTHIA with only these two nuisance parameters would have sufficient freedom to describe $p_T^W$ sufficiently accurately in a real measurement of $m_W$, and they are unlikely to accurately reflect the residual perturbative uncertainties in state-of-the-art calculations of $p_T^W$. It is nonetheless interesting to consider the $p_T^\mu$ fit performance with this simplified setup, with the expectation that in a real measurement of $m_W$ a tool with higher formal accuracy would be used in place of PYTHIA, and the nuisance parameters used in the $p_T^\mu$ fit would be chosen to – as far as possible – reflect the residual uncertainty on $p_T^W$ after, for example, tuning using $p_T^Z$ and other data.

The possibility of determining these parameters directly from the W boson data is an attractive one, as it could allow a measurement of $m_W$ to reduce its sensitivity to imperfectly modelled differences between $W^\pm$ and $Z^0$ production, such as heavy quark effects [33, 34] and flavour-dependent parton transverse momenta [35], and avoid constraining nuisance parameters to values determined using measurements of $Z^0$ production. The differences between $W^\pm$ and $Z^0$ production, and the associated uncertainties in a measurement of $m_W$, were extensively studied by the ATLAS collaboration [36].

## 2    Simulation of W production and reweighting

Monte Carlo events of the inclusive process $pp \rightarrow W \rightarrow \mu\nu$, at a centre-of-mass energy $\sqrt{s} = 13\,\text{TeV}$, are generated using PYTHIA [28] version 8.235 and the `NNPDF23_lo_as_0130_qed` PDF set [37]. Samples are generated for a $4 \times 4$ grid of different parton-shower $\alpha_s$ and $k_T^{\text{intr.}}$ parameters. These are the two parameters, in PYTHIA, that most strongly affect the $p_T^W$ distribution. Their precise definitions, and the ranges over which they are varied, are detailed in Appendix A. For this study around $1.7 \times 10^8$ events are produced at each of the 16 grid points, corresponding to around three times the expected yields given in Ref. [7] for the $6\,\text{fb}^{-1}$ Run 2 dataset recorded by LHCb. The effect of these parameter variations on the $p_T^W$ distribution is shown in Fig. 1.

These events are reweighted to different values of $m_W$ using a relativistic Breit-Wigner function with mass-dependent width,

$$\left((m^2 - m_W^2)^2 + m^4\Gamma_W^2/m_W^2\right)^{-1},$$

where the W boson width, $\Gamma_W$, is fixed to its nominal value and $m$ denotes the W propagator mass. Reweighting to arbitrary values of the nuisance parameters $\alpha_s$ and $k_T^{\text{intr.}}$

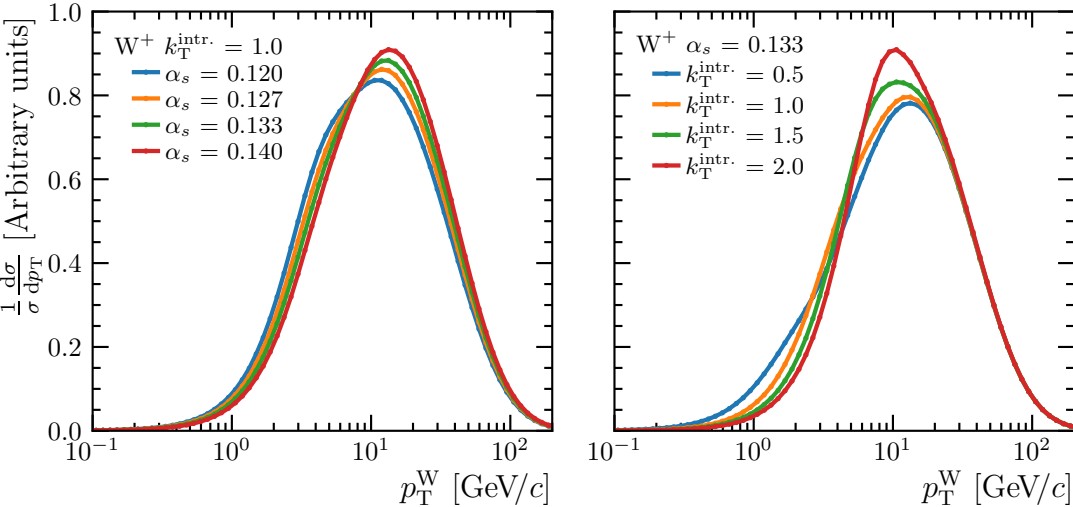

Figure 1: Illustration of variations in the $W^+$ boson $p_T$ spectrum corresponding to variations in the $\alpha_s$ (left) and $k_T^{\text{intr.}}$ (right) nuisance parameters. No kinematic requirements have been placed on the $W^+$ decay products. The equivalent distributions for the $W^-$, which qualitatively look very similar, are shown in Appendix B.

is based on three-dimensional histograms of the W propagator mass[1], rapidity and $p_T$ that have been populated with the events from each point on the $4 \times 4$ grid. These are interpolated to the desired values of $\alpha_s$ and $k_T^{\text{intr.}}$ using a two-dimensional cubic spline.

# 3  Fitting method

The values of $m_W$ and the nuisance parameters $\alpha_s$ and $k_T^{\text{intr.}}$ are determined using a binned maximum likelihood fit to $p_T^\mu$. In each fit, the signal shape is described using Monte Carlo template events, which are reweighted on the fly as the values of $m_W$, $\alpha_s$ and $k_T^{\text{intr.}}$ vary. The Beeston-Barlow "lite" prescription [38, 39] is used to account for the finite Monte Carlo statistics in the signal templates. An example fit is shown in Fig. 2, where all three of $m_W$, $\alpha_s$ and $k_T^{\text{intr.}}$ are allowed to vary, and the pseudodata statistics mirror Ref. [7].

The studies in this paper are based on pseudodata fits, where in each fit the pseudodata are drawn from one point on the $4 \times 4$ grid, and the signal templates are based on events from a different point on the grid. The pairs of grid points are chosen according to the scheme illustrated in Fig. 3. The number of independent pseudodata fits that can be run, therefore, scales inversely with the desired statistics in each fit. The baseline configuration scales down the statistics assumed in Ref. [7] by a factor of four in order to boost the number of independent pseudoexperiments that can be run. The number of signal template events is limited to a maximum of ten times the pseudodata yield.

---

[1]As reported in the PYTHIA event history.

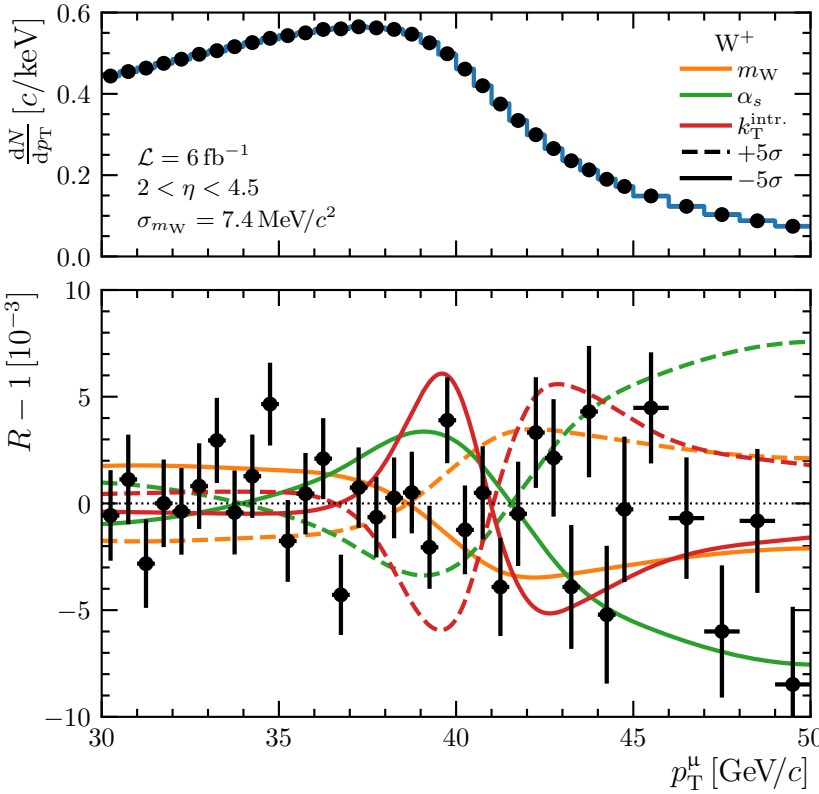

Figure 2: Illustrative fit result from a simultaneous fit to the $W^+$ (shown) and $W^-$ (see Appendix B) $p_T^\mu$ distributions. This fit assumes the statistics and fiducial region of Ref. [7]. The lower panel shows $R - 1$, where $R$ denotes the various curves divided by the best-fit distribution. The coloured curves illustrate the variation in the template distribution when the parameter given in the legend is varied by $\pm 5\sigma$, where $\sigma$ is the uncertainty reported by the fit.

# 4   Pseudoexperiment results

The baseline configuration for the results in this paper is to adopt the $30 < p_T^\mu < 50\,\mathrm{GeV}/c$ and $2 < \eta < 4.5$ kinematic region chosen by Ref. [7], with pseudodata statistics reduced by a factor four with respect to that study as noted in Sect. 3. The baseline choice is to allow three physical parameters to float in each fit: $m_W$, $\alpha_s$ and $k_T^{\mathrm{intr.}}$. Various changes to the kinematic region and the choice of free parameters are also explored.

This baseline configuration produces unbiased results with good statistical coverage, as illustrated by Figs. 4, 5 and 6, where results from every point on the $4 \times 4$ grid are combined. With the baseline configuration and available yields there are 192 independent pseudodatasets, of which 191 survive minimal quality requirements. The uncertainties are found to be well approximated by symmetric Gaussian behaviour. For brevity, in the rest of the paper, when we consider departures from the baseline configuration, such distributions are summarised by their means and widths. For example, variations in the number of fit parameters are shown in Fig. 7, indicating that the fit procedure performs well under all considered variations, the most of extreme of which is to simultaneously fit $m_W$, and separate values of the nuisance parameters $\alpha_s$ and $k_T^{\mathrm{intr.}}$ for each W boson charge. Further results illustrating the stability of the fit procedure are given in Appendix C.

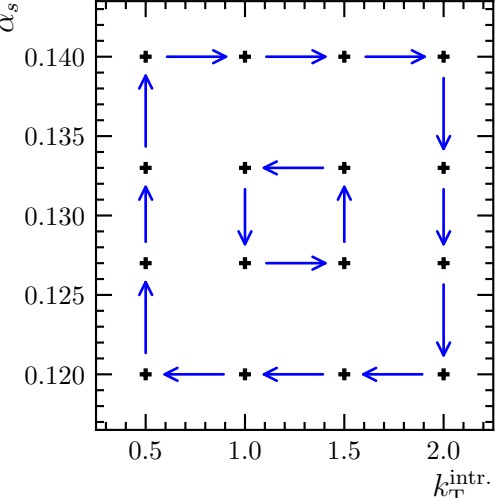

Figure 3: Illustration of relationship between the grid point from which pseudodata is drawn (arrow head) and that from which the signal template events are taken (arrow tail). The choice of nearby points maximises the statistical power of the template events.

Figure 4: Normalised residuals of $m_W$ using the baseline fit configuration of Sect. 3. The mean and spread of the unbinned data, and a corresponding Gaussian curve, are overlaid.

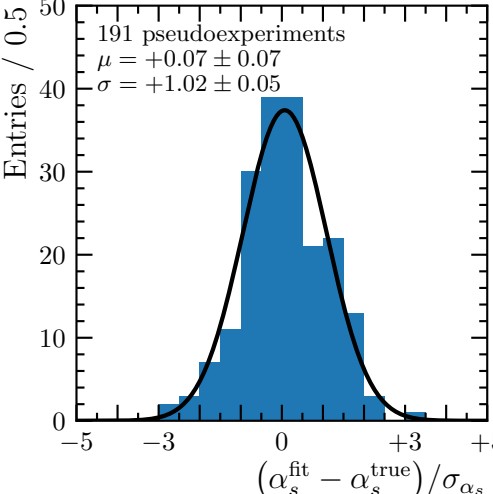

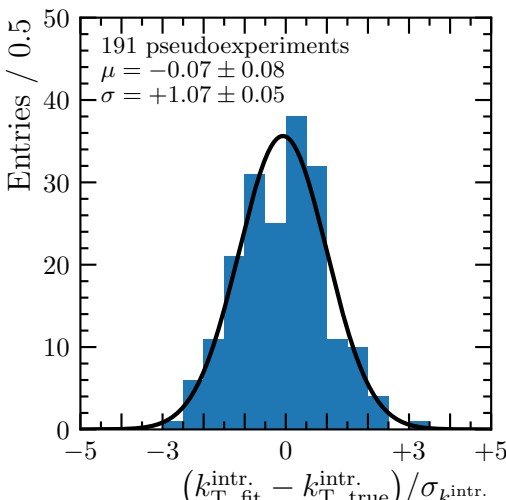

Figure 5: Normalised residuals of $\alpha_s$ using the baseline fit configuration of Sect. 3. The mean and spread of the unbinned data, and a corresponding Gaussian curve, are overlaid.

Figure 6: Normalised residuals of $k_T^{\text{intr.}}$ using the baseline fit configuration of Sect. 3. The mean and spread of the unbinned data, and a corresponding Gaussian curve, are overlaid.

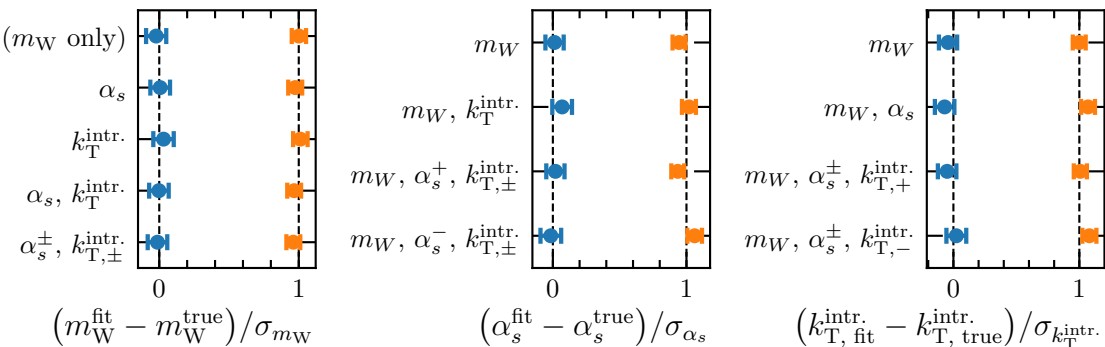

Figure 7: Summary of the mean (blue) and width (orange) of the normalised residual distributions obtained from pseudoexperiments with different sets of parameters allowed to vary. The $y$ axis labels indicate which parameters are free to vary in addition to the parameter shown on the $x$ axis. Where signs appear in parameter names, such as $\alpha_s^+$ and $k_{\mathrm{T},-}^{\mathrm{intr.}}$, this indicates that the parameter may take different values for $\mathrm{W}^+$ and $\mathrm{W}^-$, and $\alpha_s^\pm$ is shorthand for $\alpha_s^+$, $\alpha_s^-$ and so on. The baseline configuration corresponds to the fourth row in the leftmost figure, and the second row in the centre and right figures.

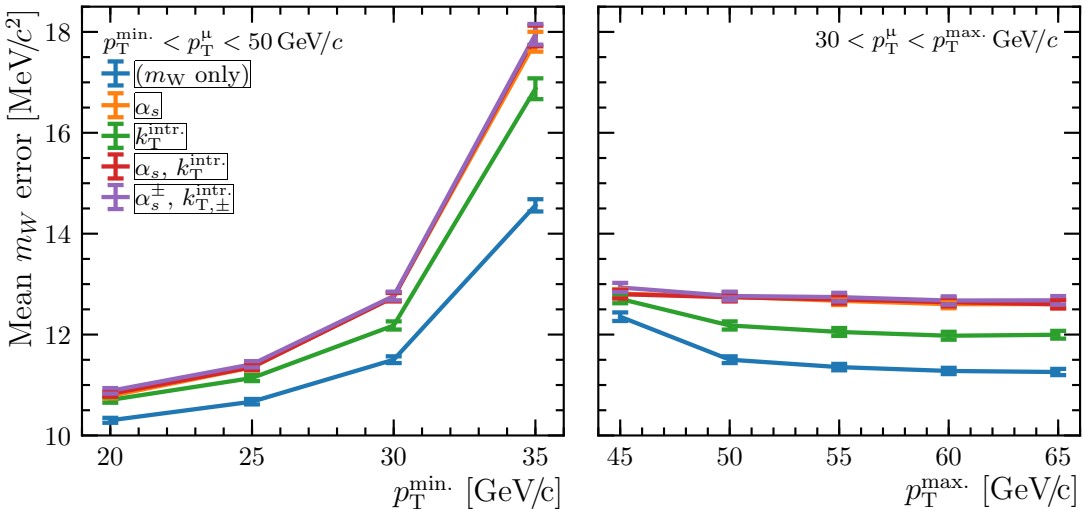

Figure 8: Variation of the statistical uncertainty on $m_\mathrm{W}$ obtained from several fit configurations, illustrated as a function of the fit range in $p_\mathrm{T}^\mu$. As in Fig. 7, the legend indicates which parameters are free to vary in addition to the parameter, $m_\mathrm{W}$, shown on the $y$ axis. The baseline configuration corresponds to the red curve at $p_\mathrm{T}^{\mathrm{min.}}$ $(p_\mathrm{T}^{\mathrm{max.}}) = 30 \ (50) \,\mathrm{GeV}/c$ in the left (right) figure.

Having demonstrated that the pseudoexperiment setup performs well, it is interesting to explore how the fit results depend on choices such as the $p_\mathrm{T}^\mu$ fit range and the number of freely varying nuisance parameters. One such study is shown in Fig. 8, which shows the the average statistical uncertainty on $m_\mathrm{W}$ for several choices of fit range and fit parameters. This shows that the proposed method incurs only a modest degradation in statistical precision with respect to the simplest $m_\mathrm{W}$-only fit configuration, and interestingly that allowing the $\mathrm{W}^+$ and $\mathrm{W}^-$ to each take their own value of the two nuisance parameters has a negligible effect.

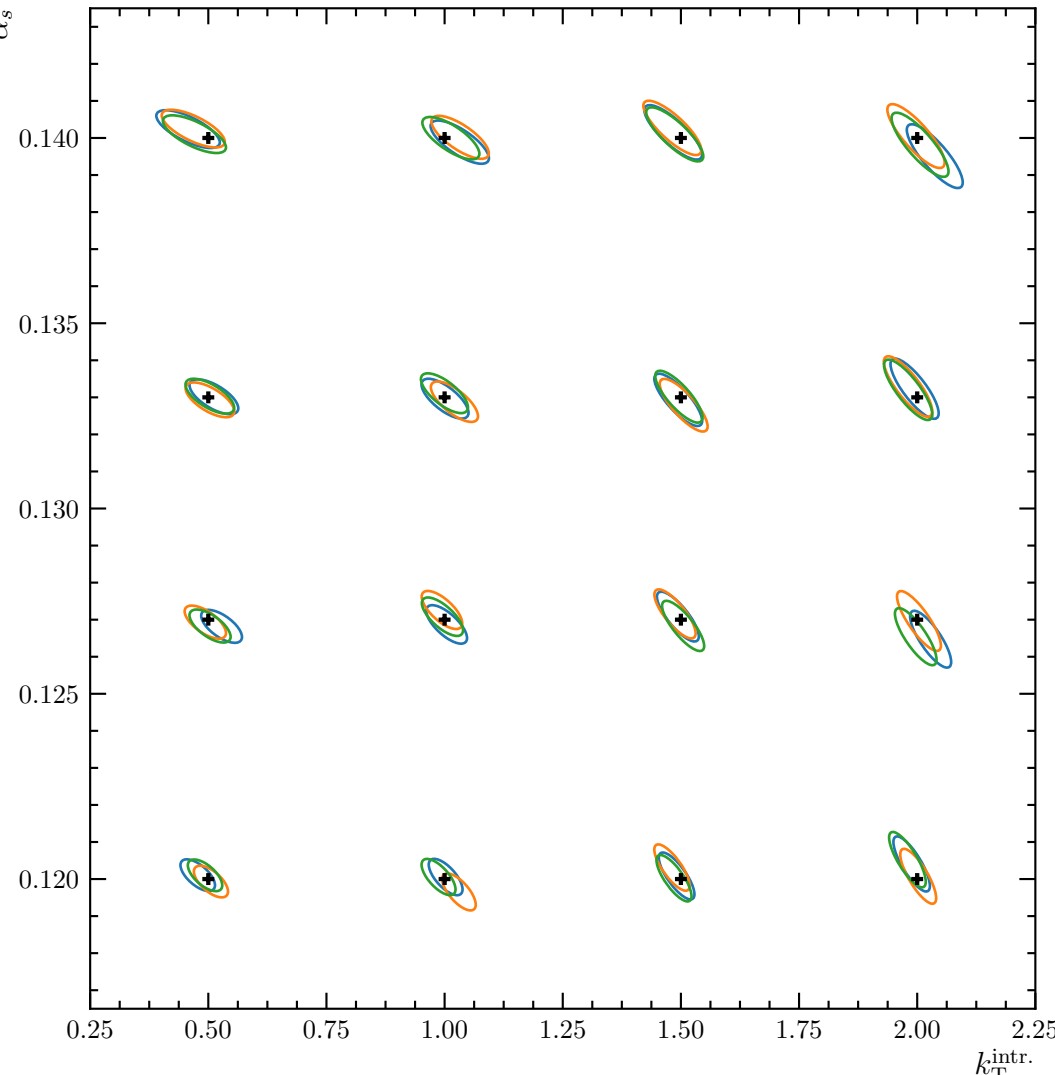

Figure 9: Illustration of the $4 \times 4$ interpolation grid and the Gaussian $3\sigma$ error ellipses obtained from each pseudoexperiment, showing the significant anti-correlation between the two nuisance parameters. Here the pseudoexperiments correspond to the event yields of Ref. [7], *i.e.* they are a factor four higher than the baseline configuration. The different colours have no particular meaning, and simply serve to differentiate the different pseudoexperiment results.

The two nuisance parameters chosen for this study exhibit a significant anti-correlation, as might be expected from Fig. 2, which is illustrated in Fig. 9. The fit performance distributions already shown indicate that this is not a major problem, but it can also be seen in Fig. 10 that increasing the upper $p_T^\mu$ limit would reduce this correlation, as the large $p_T^\mu$ range is principally sensitive to $\alpha_s$. The extent to which the proposed method can disentangle $m_W$ from the other QCD nuisance parameters can also be probed by examining the global correlation coefficient of $m_W$ [40]. This is defined as the correlation between $m_W$ and the linear combination of all other fit parameters that it is most strongly correlated with, and it is shown for $m_W$ in Fig. 11. It can be seen that reducing $p_T^{\text{min.}}$ tends to reduce the degeneracy of the fit parameters.

It is also confirmed that adopting the event yields of Ref. [7], *i.e.* increasing those

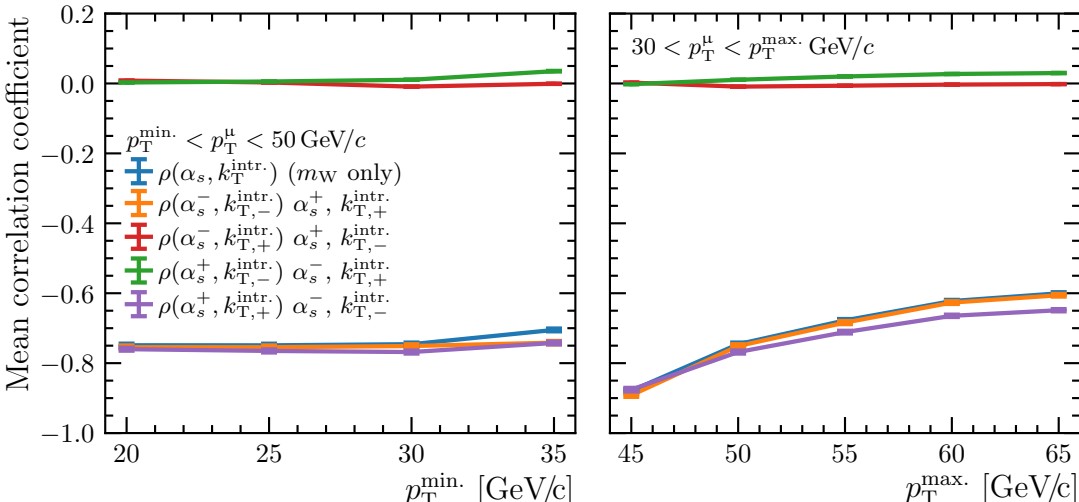

Figure 10: Variation of the $\alpha_s$–$k_\mathrm{T}^{\mathrm{intr.}}$ correlation obtained from several fit configurations, illustrated as a function of the fit range in $p_\mathrm{T}^\mu$. The meaning of the superscript charges is defined in Fig. 7 and the legend entries are described in Fig. 8.

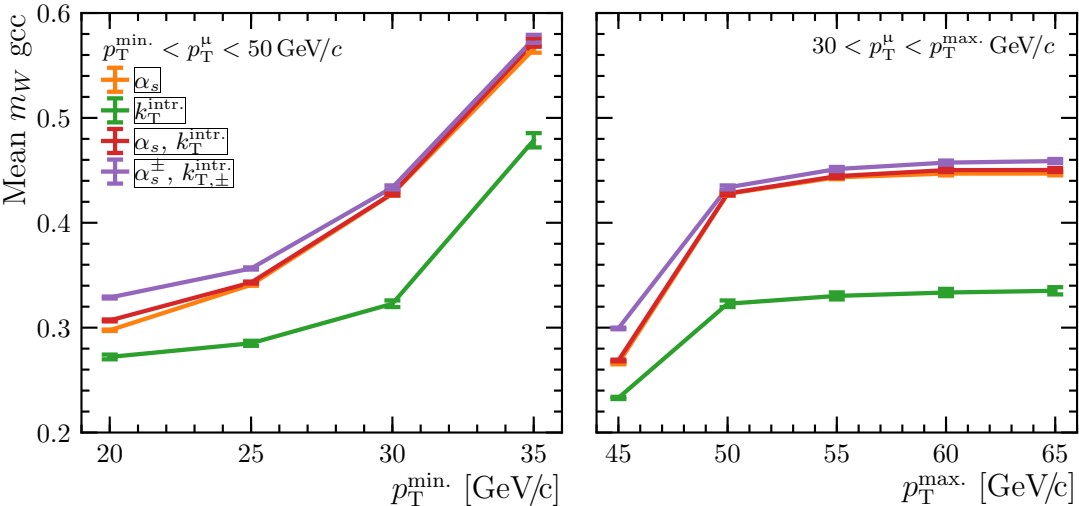

Figure 11: Variation of the global correlation coefficient (gcc) of $m_\mathrm{W}$ obtained from several fit configurations, illustrated as a function of the fit range in $p_\mathrm{T}^\mu$. The meaning of the superscript charges is defined in Fig. 7 and the legend entries are described in Fig. 8. Note that, as reported in Figs. 17 and 18, sensitivity to $\alpha_s$ and $k_\mathrm{T}^{\mathrm{intr.}}$ deteriorates significantly at $p_\mathrm{T}^{\mathrm{max.}} = 45\,\mathrm{GeV}/c$.

of the baseline configuration by a factor four, does not introduce any bias or coverage problems, and it is this higher-statistics configuration that is illustrated in Figs. 2 and 14.

Several additional figures showing the various parameter uncertainties, their correlations and the variation of these quantities with different fit configurations are included in Appendix C.

# 5 Conclusions

We have demonstrated that it is possible to simultaneously determine both the W boson mass, $m_W$, and nuisance parameters relating to its $p_T$ spectrum using a fit to the $p_T^\mu$ spectrum with only a small inflation of the statistical uncertainty on $m_W$. We find that, for the specific parameters that were chosen to illustrate the technique, the simultaneous fit is well-behaved and that for most reasonable choices of the $p_T^\mu$ fit range the fits have little trouble disentangling variations in $m_W$ from those in the $p_T^W$ model. The study considers variations of the nuisance parameters that correspond to variations in the $p_T^W$ spectrum that are large compared to the uncertainty of state-of-the-art predictions, indicating that the proposed technique is sufficiently powerful to enable a precise measurement of $m_W$.

In an actual measurement of $m_W$ it would, of course, be preferable to apply the same technique using predictions from tools that contain higher order electroweak and QCD corrections, which naturally leads to the question of what parameters can legitimately be varied in this case. The examples that have been shown to work well with PYTHIA in this study could provide a useful template: even in the more accurate calculations it should be possible to identify a $k_T^{intr}$-like nonperturbative smearing, and to vary the strong coupling constant, but other choices may prove to be optimal for different tools. A larger number of nuisance parameters could also be varied simultaneously, if this was well motivated for a particular tool; the implementation used in this paper in theory has no upper limit, but in practice it is limited to varying a maximum of 3–4 parameters in addition to $m_W$.

It will also be interesting to explore how this method can be combined with the techniques explored in Ref. [7] for reducing PDF uncertainties using in-situ constraints, and it is important to verify that the inclusion of realistic levels of QCD and electroweak backgrounds does not adversely affect the performance of the method.

In summary, the proposed technique performs well using pseudodata generated with PYTHIA, and appears to provide a possible route to a precise measurement of $m_W$ that is less reliant on accurate modelling of the differences between W and $Z^0$ boson production.

# Acknowledgements

We thank W. Barter, M. Charles, S. Farry, R. Hunter, M. Pili, F. Tackmann and A. Vicini for their helpful comments and suggestions during the preparation of this manuscript. OL thanks the CERN LBD group for their support during the initial stages of this work, and MV thanks the Science and Technologies Facilities Council for their support through an Ernest Rutherford Fellowship.

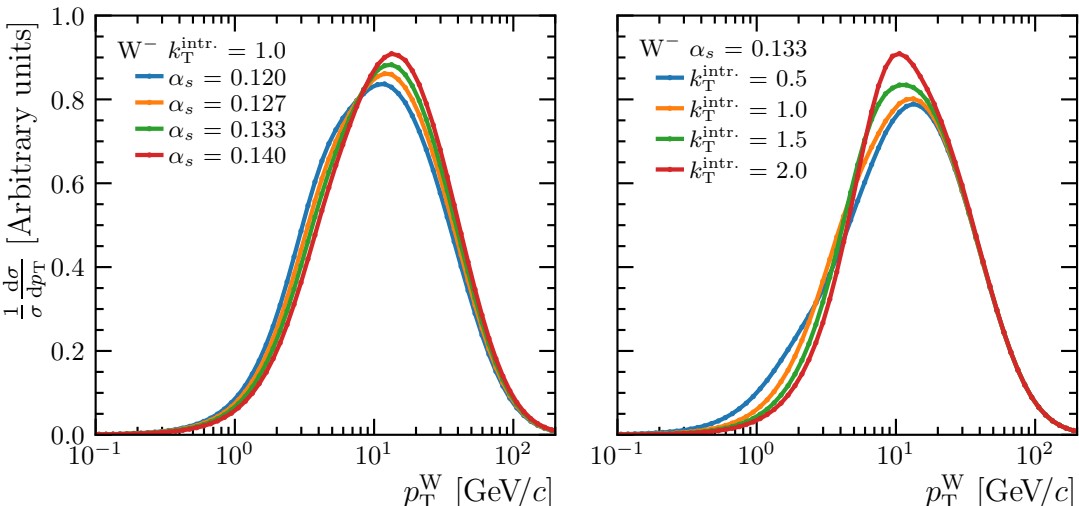

Figure 12: Illustration of variations in the W$^-$ boson $p_T$ spectrum corresponding to variations in the $\alpha_s$ (left) and $k_T^{\text{intr.}}$ (right) nuisance parameters. No kinematic requirements have been placed on the W$^-$ decay products. This is the W$^-$ analogue of the W$^+$ distributions shown in Fig. 1.

# Appendices

Appendix A provides additional detail regarding the PYTHIA configuration used throughout this study, while Appendix B includes the W$^-$ counterpart of the illustrative W$^+$ fit distribution shown in the main text. Finally, Appendix C includes additional results from the ensemble of pseudoexperiments.

# A  Pythia tuning parameters

The quantity $\alpha_s$ used throughout this paper refers to the PYTHIA configuration options `TimeShower:alphaSvalue` and `SpaceShower:alphaSvalue`, while the quantity $k_T^{\text{intr.}}$ is a scale factor applied to the configuration options

$$\texttt{BeamRemnants:halfScaleForKT} = 1.5 \times k_T^{\text{intr.}},$$

$$\texttt{BeamRemnants:primordialKTsoft} = 0.9 \times k_T^{\text{intr.}},$$

$$\texttt{BeamRemnants:primordialKThard} = 1.8 \times k_T^{\text{intr.}}.$$

The $4 \times 4$ grid consists of $\alpha_s \in \{0.120, 0.127, 0.133, 0.140\}$ and $k_T^{\text{intr.}} \in \{0.5, 1.0, 1.5, 2.0\}$. With the exception of these parameters, the default tuning of PYTHIA 8.235 is used.

# B  Additional kinematic distributions

This section contains Fig. 12, which is the W$^-$ counterpart to the W$^+$ boson $p_T$ distributions shown in Fig. 1, Fig. 13, which is a more verbose analogue to Fig. 1, and Fig. 14, which is the W$^-$ counterpart of the W$^+$ fit, Fig. 2, shown in the main body of the paper.

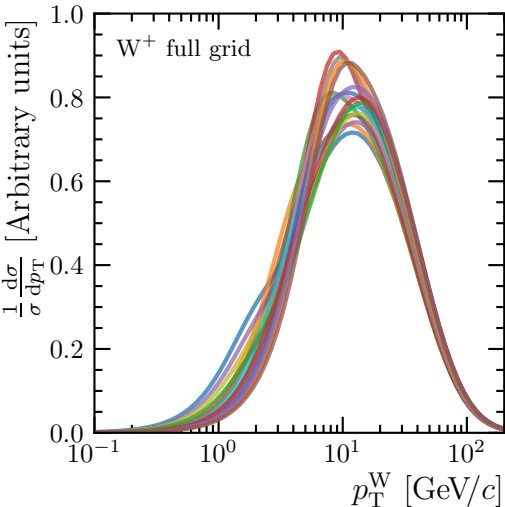

Figure 13: Illustration of variations in the $W^+$ boson $p_T$ spectrum corresponding to all 16 points on the $4 \times 4$ grid of $\alpha_s$ and $k_T^{\text{intr.}}$ values.

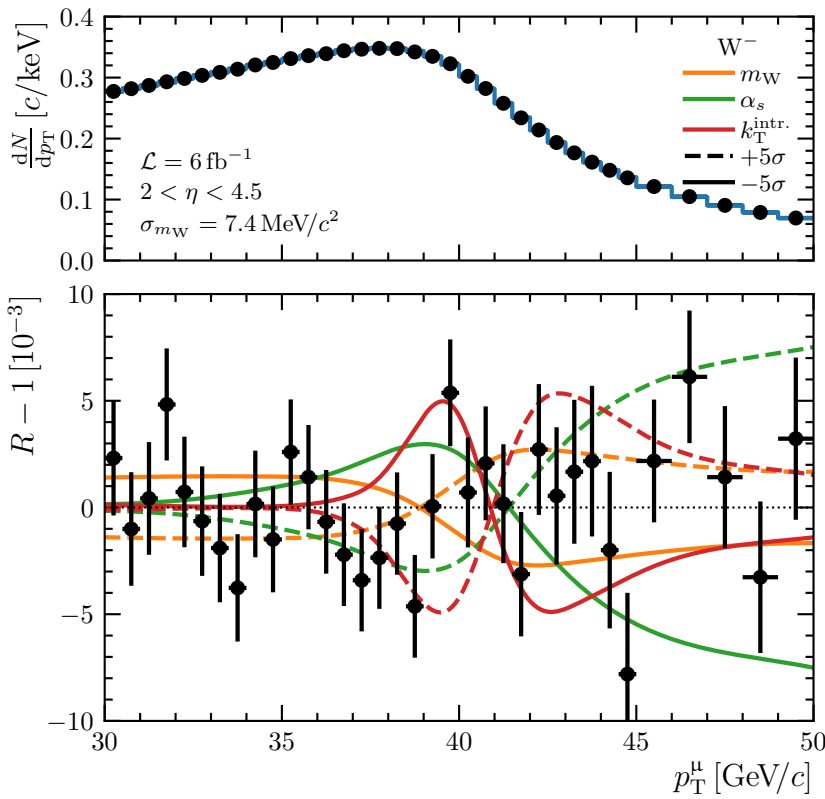

Figure 14: Illustrative fit result from a simultaneous fit to the $W^+$ (see Sect. 3) and $W^-$ (shown) $p_T^\mu$ distributions. Further information is given alongside Fig. 2.

# C  Additional pseudoexperiment results

This section contains additional results showing the stability of the fit procedure with respect to different departures from the baseline configuration. Figure 15 shows that the fit procedure is reasonably stable as the absolute values of the nuisance parameters vary across the $4 \times 4$ grid of $\alpha_s$ and $k_T^{\text{intr.}}$ values. Figure 16 shows the fit procedure remains stable when the allowed range of the muon transverse momentum, $p_T^{\mu}$, is varied. Figures 17 and 18 show the dependence of the uncertainties with which $\alpha_s$ and $k_T^{\text{intr.}}$ are determined on the fit configuration, while Figs. 19 and 20 show the global correlation coefficients of these parameters. Finally, Figs. 21 and 22 show the correlation coefficients between $m_W$ and $\alpha_s$ and $k_T^{\text{intr.}}$.

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
