# Peer review of "Simultaneously determining the $W^{\pm}$ boson mass and parton shower model parameters"

_SciPost Physics_

## Round 2 · Referee Report · Anonymous (Referee 1) · 2019-12-19

Strengths

the paper “Simultaneously determining the W? boson mass and parton shower model parameters”
by O. Lupton and M. Vesterinen is a study of fits of the muon pt spectrum in W decays from Pythia generated pseudo-data in a kinematic region similar to the acceptance of the LHCb experiment . They fit simultaneously the W mass and two Pythia parameters , relevant for shaping the W pt distribution. The main result of the paper is that the correlation of the W mass with these two parameters is in the 30\%-60\% region depending on the rage of the fit.

I think the paper is technically sound and reports a wealth of information about those fits: correlations among the parameters, dependence on the fit range.

Weaknesses

The question that comes natural is: what do we learn from these fits ?
\begin{enumerate}
\item The abstract says “that there is little trouble in disentangling variations in the muon pt spectrum due to Mw from those due to the W boson pt model”. This is not surprising: Mw generates (almost) a singularity in the muon spectrum in the W reference frame, which is then smeared by the W Pt. The two parameters in the study cannot be *very* correlated with Mw unless one restrict the study of the spectrum in a small region near Mw/2.
\item The author themselves recognise that the proposed method is not suited for a W mass measurement with small systematic uncertainty , see second paragraph at page 2. And they correctly do not provide an estimate of the systematic uncertainty on Mw following from the method that they propose.
\end{enumerate}

Report

Dear Editor,

the paper “Simultaneously determining the W? boson mass and parton shower model parameters”
by O. Lupton and M. Vesterinen is a study of fits of the muon pt spectrum in W decays from Pythia generated pseudo-data in a kinematic region similar to the acceptance of the LHCb experiment . They fit simultaneously the W mass and two Pythia parameters , relevant for shaping the W pt distribution. The main result of the paper is that the correlation of the W mass with these two parameters is in the 30\%-60\% region depending on the rage of the fit.

I think the paper is technically sound and reports a wealth of information about those fits: correlations among the parameters, dependence on the fit range. The question that comes natural is: what do we learn from these fits ?
\begin{enumerate}
\item The abstract says “that there is little trouble in disentangling variations in the muon pt spectrum due to Mw from those due to the W boson pt model”. This is not surprising: Mw generates (almost) a singularity in the muon spectrum in the W reference frame, which is then smeared by the W Pt. The two parameters in the study cannot be *very* correlated with Mw unless one restrict the study of the spectrum in a small region near Mw/2.

\item The author themselves recognise that the proposed method is not suited for a W mass measurement with small systematic uncertainty , see second paragraph at page 2. And they correctly do not provide an estimate of the systematic uncertainty on Mw following from the method that they propose.
\end{enumerate}

The important question is - in my opinion - to which precision the W pt spectrum can be modelled from the proposed variations, and in general using parton-shower algorithms ? This question is not addressed in the paper. And I think it cannot be answered in a robust way. Another interesting and correlated question- not addressed in the paper - is : how well do we need to know the W pt spectrum in order to have a systematic uncertainty on W mass smaller than say 10 MeV ? There is a generic statement at page 1, however would be nice to see if there are peculiarities in this special forward kinematic region.

For the reasons mentioned above I find that this paper adds little information to the public discussion on the W mass measurement.

In the conclusions the authors say that “In an actual measurement of mW it would, of course, be preferable to apply the same technique (eg simultaneous fit of Mw and some nuisance parameters) using predictions from tools that contain higher order electroweak and QCD corrections, which naturally leads to the question of what parameters can legitimately be varied in this case” . This is a more interesting subject, but it is not discussed in the paper.

---

## Round 2 · Referee Report · Anonymous (Referee 2) · 2020-1-10

Strengths

  1. Good background, context and motivation for the studies performed.
  2. Technically/statistically sound procedure with comprehensive description of the methodology.
  3. Good visualisation of the results.

Weaknesses

  1. Extremely simplified model.
  2. Lack of quantitative results on the uncertainty in the W transverse momentum distribution and its relationship to the additional uncertainty on the extracted W mass.

Report

The issue which is explored is highly relevant to measurements of the W mass at the LHC, and the studies which are performed are technically sound and well described. The model which is used is however extremely simplified and very far from what would be used in a real measurement, though these limitations are accurately acknowledged in the paper. It is not straightforward to compare the in-situ constraints achieved by the fit with this toy model to existing theoretical predictions or to understand how this is related quantitatively to the residual uncertainty on the W mass.

Requested changes

The paper would benefit from both a visualisation and quantification of the post-fit uncertainty on the W transverse momentum distribution itself in the different scenarios, and how this relates to the residual uncertainty on the extracted W mass.

I would suggest at least one illustrative plot showing the W transverse momentum distribution with postfit uncertainty bands for one of the scenarios, possibly with additional lines showing variations from individual parameters and/or post-fit eigenvectors to visualise the correlation structure.

This could possibly also be extended to plots in the style of Fig. 8 showing the relationship between the uncertainty on the W pT distribution at one or more values vs the fit range in muon pT, which could then be compared to the corresponding uncertainty on mW itself.

---

## Round 2 · Referee Report · Anonymous (Referee 3) · 2020-2-18

Strengths

1 The paper studies an important open issue, specifically the potential for constraining theory uncertainties in the W boson mass measurement at LHCb using the pTl distribution in W boson events.

2 The results are self-contained in that the effects of both non-perturbative and perturbative uncertainties on the W boson pT are shown, and the statistical impact on the measurement is evaluated.

3 The results are qualitatively as expected and the methodology appears to be scientifically sound.

Weaknesses

1 The scope is limited, leaving open the crucial question of how much the proposed methodology would reduce the corresponding theory uncertainties, and more importantly, the overall W boson mass uncertainty.

Report

The submitted paper details the statistical sensitivity of the LHCb experiment to the measurement of the W boson mass (MW) and non-perturbative and perturbative parameters affecting W boson production, through a multiparameter fit to the muon pT distribution. The paper shows the statistical correlations of the parameters, the impact of variations of each on the distribution, and the reduction in statistical precision on MW from the multiparameter fit. Modelling the pTW distribution is a crucial component of the MW measurement at LHCb, and any progress in that regard is a welcome and important development.

The method proposed by the authors has not been used before, which can be viewed positively (it is a novel strategy) or negatively (it has been consciously avoided). The strategy of previous experiments has been to precisely determine the Z boson pT to constrain both perturbative and non-perturbative parameters that are expected to be common to W and Z boson production. For example, ATLAS use a similar PYTHIA model to that studied by the authors, and explicitly take additional uncertainties for expected differences between W and Z production due to the flavour dependence of initial quark momentum distributions. The obvious question is what the LHCb uncertainties would be using a similar strategy (i.e. constraining the PYTHIA parameters with Z data and including residual theory uncertainties), and how would the fit to the pTl distribution affect these uncertainties? Providing this information would crucially quantify the relative value of a multiparameter fit to the pTl distribution.

The authors further note that a more advanced QCD calculation may be required to accurately model the LHCb pTW distribution. This raises the question of a potential modelling systematic uncertainty associated with the simultaneous fit to the pTl distribution. How would a different calculation with different parameters, e.g. a fixed-order + resummed calculation with scale variations, alter the distribution? A full study would compare the impact of these variations as well as variations on the PYTHIA parameters. Since this would involve the use of at least one additional generator, it is perhaps outside the scope of this study, but the potential for an additional pTW modelling uncertainty affecting the MW extraction should be noted. Furthermore, the pTl distribution receives many experimental uncertainties (e.g. resolutions and pT-dependent efficiencies) that will affect the multiparameter fit, and should also be mentioned (e.g. the resolution uncertainty should affect the alphaS uncertainty).

In summary, the paper discusses the important open question of pTW modelling and its impact on the MW measurement at LHCb, adding value to the ongoing investigations. However, the discussion lacks the context of the expected uncertainty using existing methodology. In my opinion this limits the paper's value.

Requested changes

As stated above, I recommend discussing the uncertainty on the pTW modelling parameters constrained with Z boson data at LHCb (and possibly ATLAS & CMS), and the expected improvement from the pTl multiparameter fit. Also please discuss the systematic uncertainties affecting the parameter extraction (modelling and experimental).

I list here a couple of other minor comments and requests:

Section 1: Please clarify "intrinsic kT cut off parameter" -- probably mean "cutoff"? This parameter name could be given in Appendix A.

Section 2: Optionally add the impact of these parameters on pTl (this is addressed to some extent in Fig. 2)

Appendix B: "more verbose analogue"? Maybe more "detailed" or more "complete"?

Appendix C: The means and widths in Figure 15 have large deviations that seem unlikely to be purely from fluctuations -- can you check the results in this plot?

---

## Round 2 · Referee Report · Anonymous (Referee 4) · 2020-3-6

Strengths

1 - The work performed is clearly described
2 - The study is well motivated (investigation of the effect of a simultaneous fit to the W Mass and nuisance parameters on the measured W Mass)

Weaknesses

1 - The study does not address the questions raised in the motivation to a level that suggests it is mature for publication. In particular citation [7] (co-authored by the authors of this paper), shows that the PDF's can be constrained from measurement of the muon $p_T$ distribution in a similar kinematic range. This must mean that any attempts to constrain the Monte-Carlo parameter values in addition to $m_W$ using the same distribution will underestimate the uncertainty due to the combined fit unless the effects of the PDF uncertainties are included.

2 - Reliance on Pythia introduces an unrealistic knowledge of the nuisance parameters which cannot be reproduced when working with data. The study will be more robust if the signal pseudo-experiment data are determined from a different generator (ResBos, or PowHeg+Pythia). The $\chi^2/NDF$ of the best fit muon $p_T$ distribution and the corresponding test of determined W Mass will be closer to an implementation with data.

3 - Ref [4] mentions the factorization scale and the heavy quark masses as the parameters which are not constrained by fits to the Z Boson $p_T$ distribution. These parameters are not constrained by this study. The current note doesn't contrast their method to constraining the same pythia parameters that they consider with the Zboson $p_T$.

4 - The actual uncertainties on the non $m_W$ parameters are not shown in comparison to the determination in reference [32]

Report

Measurement of the W Boson mass ($m_W$) is of importance and interest to the community, and any advances which would help to reduce the associated uncertainties is very welcome.

In this study the authors investigate the impact on the W Mass uncertainty from a measurement at high $\eta$ (within the LHCb acceptance). This is motivated by previous studies which show a reduction in $\Delta m_W$ (particularly the contribution from PDF's) when central and forward measurements are combined.

In particular the authors perform a simultaneous fit to the lepton produced in W boson decay to determine the W Mass and two parameters within Pythia which control the boson $p_T$ distributions. This is motivated as being an attempt to reduce the systematic uncertainty from modelling of differences in the Z and W boson $p_T$ distributions. In my opinion the study is not yet complete as the true effect on expanding the number of fit parameters has not been fully presented. The effects due to PDF uncertainties, a different generator model (which calculates the boson $p_T$ using a different model), and some estimation of experimental effects (lepton resolution, and to a lesser extent an event selection simulation for LHCb) are conspicuously absent.

Most of the studies which propose measurements of the $m_W$ at LHCb refer to a arXiv:1408.4354v2 which was a determination of the forward W Boson cross section at LHCb. A large source of the uncertainty in that measurement is the shape uncertainty in the $\mu_{p_T}$ distribution. which combines variations in the signal and background process shapes. It's not clear to me how much this nullifies the results in this note.

With this in mind I am of the opinion that the work as presented does not meet the publication criteria of Scipost Physics.
https://scipost.org/SciPostPhys/about#criteria

With a little more work I believe it may be, and I encourage the authors to continue to expand their study.

Requested changes

1 - Estimate results with PDF variations included

2 - Compare to results with pythia parameters fixed by fit to the $Z \rightarrow \mu \mu \ p_T $ distribution.

3 - Possibly use a different generator model for the signal pseudodata.

---

## Editorial Decision

awaiting_resubmission